# Statistical methods to disentangle genetic effects influencing infertility and early fetal viability with a genome-wide application

Siri N. Skodvin[1,2]*, Miriam Gjerdevik[1,3], Julia Romanowska[1,2], Siri E. Håberg[1,2], Alexandra Havdahl[4,5], Rolv Terje Lie[1,2], Astanand Jugessur[1,2], Håkon K. Gjessing[1,2]

1 Centre for Fertility and Health, Norwegian Institute of Public Health, Oslo, Norway, 2 Department of Global Public Health and Primary Care, University of Bergen, Bergen, Norway, 3 Department of Computer Science, Electrical Engineering and Mathematical Sciences, Western Norway University of Applied Sciences, Bergen, Norway, 4 PsychGen Centre for Genetic Epidemiology and Mental Health, Norwegian Institute of Public Health, Oslo, Norway, 5 Nic Waals Institute, Lovisenberg Diaconal Hospital, Oslo Norway

* sirinaerland.skodvin@fhi.no

## Abstract

Genetic selection occurs at multiple stages before and during pregnancy. While parental genomes influence the probability of fertilization, the fetal genome, once established, plays a critical role in early fetal survival. However, when estimated separately, parental and fetal genetic effects may confound each other. To address this, we developed an extension of the case-parent triad design to jointly estimate the genetic contributions of the parents and the fetus. Our approach considers all offspring as carriers of the trait "fetal survival''. As use of assisted reproductive technology (ART) usually reflects fertility issues, we performed separate analyses on non-ART and ART family units, hypothesizing that parental and fetal effects differ between these groups. In the Norwegian Mother, Father, and Child Cohort Study, we had access to genotypes for approximately 43,000 family triads and dyads, including 1,336 offspring conceived through ART. In the non-ART sample, we identified genome-wide significant fetal effects on fetal survival for SNPs within regions harboring genes relevant to infertility and fetal development, such as *MDC1*, *MICB*, *HCP5*, and *NOTCH4*. These effects remained significant after adjusting for parental interaction effects, confirming their origin as fetal effects. When we replicated the analysis in the ART sample, we observed partial overlap in fetal effects with those identified in the non-ART sample. Parental interaction effects were observed in both the non-ART and ART samples, but the specific genetic associations differed between the groups. Notably, several SNPs associated with parental interaction effects in the ART sample mapped to genes previously implicated in male infertility, including *ACTB*, *FSCN1*, and *RNF216*. Our findings have broad implications for understanding the genetic architecture of infertility and fetal development. To support the interpretation of our

**Data availability statement:** MoBa data are not publicly available, but can be accessed by applying to the Norwegian Institute of Public Health (https://www.fhi.no/en/ch/studies/moba). The original Haplin can be installed from the Comprehensive R Archive Network (https://CRAN.R-project.org/package= Haplin), and further information is available on the Haplin webpage (https://haplin.bitbucket.io/). Data generated are available as .csv files and .rds files on https://github.com/siriskodvin/ART_Haplin_with_parentalinteractions.

**Funding:** This research was partly funded by the Research Council of Norway (#320656) (S.E.H.) and through its Centres of Excellence funding scheme (#262700) (S.N.S., M.G., J.R., S.E.H., R.T.L., A.J., H.K.G.) and co-funded by the European Union (ERC, BIOSFER, #101071773) (S.N.S., S.E.H.). Views and opinions expressed are however those of the author(s) only and do not necessarily reflect those of the European Union or the European Research Council. Neither the European Union nor the granting authority can be held responsible for them. A.H. is supported by the Research Council of Norway (#336085), the South-Eastern Norway Regional Health Authority (#2020022), and the European Union's Horizon Europe Research and Innovation programme (FAMILY #101057529; HOMME #101142786; Marie Skłodowska-Curie grant ESSGN #101073237). This research was also partly supported by a large infrastructure grant from the Research Council of Norway to establish "Biobank Norway" (#322672) (R.T.L, A.J., H.K.G.). The funders had no role in study design, data collection and analysis, decision to publish, or preparation of the manuscript.

**Competing interests:** The authors have declared that no competing interests exist.

results, we provide detailed descriptions of the models, highlighting their strengths and limitations.

## Author summary

It is well-known that genetic factors can influence infertility. While the combined genotypes of the parents may affect fertilization success, the fetus's genotype, once established, becomes a key determinant of early fetal survival. However, separating the genetic effects of the parents from those of the fetus is challenging. Here, we present a new method to disentangle these effects by analyzing data from over 43,000 families in the Norwegian Mother, Father, and Child Cohort Study, including 1,336 children conceived through assisted reproductive technology (ART). We analyzed ART and non-ART families separately to explore potential differences in genetic effects between these two groups. Our findings show that specific fetal genetic factors are linked to early survival, including genes with known associations with infertility and fetal development. We also found that interactions between parental genotypes influence reproductive outcomes, with notable differences between ART and non-ART families. This study provides new insights into how genetic factors influence infertility and early pregnancy outcomes, thus improving our understanding of reproductive health.

## 1 Introduction

Recent estimates from the World Health Organization suggest that 17.5% of the world's population experience infertility at some point during their lifetime [1,2]. Infertility is usually defined as the inability to achieve a clinically recognized pregnancy after having tried for 12 months or more [3]. Even under optimal conditions, the maximum chance of achieving a clinically recognized pregnancy in a given cycle is around 30-40% [4]. Given that an increasing number of couples now choose to delay childbearing until later in life [5,6], which may negatively impact their reproductive success, it is not surprising that the use of assisted reproductive technology (ART) has also increased in many parts of the world [7]. Recent data from the Medical Birth Registry of Norway show that approximately 5% of all newborns in 2022 were conceived by ART [8].

Genetic effects influencing infertility may operate through a variety of biological mechanisms, including chromosomal abnormalities, monogenic disorders, or traits with complex inheritance [9–11]. Several studies have reported associations between single-nucleotide polymorphisms (SNPs) and infertility-related conditions, such as endometriosis and ovulation disorders in women and aberrant sperm characteristics in men [12–16], while others have screened for genetic variants associated with unexplained (idiopathic) infertility [17,18]. A few recent studies have highlighted the importance of considering recessive models to search for genetic effects linked with reproductive traits. For example, Ruotsalainen et al. identified SNPs in the *TBPL2*

gene with genome-wide significant recessive effects on female infertility [19]. Oddsson et al. discovered that women homozygous for a stop-gain (nonsense) variant in the *CCDC201* gene reached menopause on average nine years earlier compared to other women [20].

Most of the above studies have examined male and female reproductive traits separately. While they, along with other research, have provided significant insights into the genetic determinants of infertility, the cause remains unknown in approximately 15-30% of couples struggling to conceive [21–23]. In some of these couples, it is plausible that specific combinations of male and female germ cells lower the likelihood of fertilization. A couple's use of ART to conceive may be regarded as a proxy for their infertility. In an earlier study, we compared ART and non-ART pregnancies to identify male-female allelic interaction effects influencing a couple's ability to conceive beyond what would be expected from each partner's individual risk of infertility, revealing interaction effects with SNPs in *DNAH17*, a gene with known links to infertility [24].

Even after a successful fertilization, a substantial proportion of embryos may still perish at an early stage before the pregnancy is clinically detectable. Around 60% of all conceptuses have been estimated to be lost prior to the missed menstrual period [4,25]. Repeated early pregnancy losses may also be perceived as couple-level infertility, and parental alleles associated with infertility may in reality increase the risk of early pregnancy loss after being transmitted to the embryo. Currently, research on genetic determinants of embryonic survival is limited. However, in a study by Mayor-Olea et al., it was suggested that selection in favor of the T allele in the *MTHFR* 677C>T polymorphism may be associated with early fetal viability [26]. Recently, Arnadottir et al. provided more direct evidence that at least 1 in 136 pregnancies was lost due to a pathogenic small sequence variant, based on whole-genome sequencing of samples from fetal-parent triads affected by pregnancy loss and adult control triads [27].

To shed further light on infertility caused by early pregnancy losses, we previously investigated allele distributions in ART-conceived offspring, conditional on parental genotypes [28], and analyzed case-parent units using the R package Haplin [29–31]. If genetic influences on early pregnancy loss are more pronounced among couples who seek ART treatment, it is conceivable that the ART procedures mitigate such genetic influences. Thus, genetic variants associated with early pregnancy loss would exhibit allele frequencies that deviate from expected Mendelian parent-to-offspring transmission patterns in both non-ART and ART offspring. Under such a scenario, the frequencies of potentially deleterious alleles would also be expected to differ between non-ART and ART offspring. Indeed, we found several alleles whose frequencies in the ART offspring differed from expected proportions in two genes, *CXXC4-AS1* and *DYNLRB2-AS1* [28].

A research question that remains to be explored is how to identify whether specific genetic variants influence the probability of fertilization or embryonic survival. When parental and fetal genetic effects are analyzed separately, each type of effect may be "aliased" by the other; in other words, a parental effect could manifest as a fetal effect, and vice versa [32]. To address this analytic problem, we developed a novel approach to disentangle parental genetic interaction effects influencing fertilization success from fetal genetic effects influencing early fetal survival. Our methodology is based on log-linear modeling and builds on existing framework for case-parent triad analyses available in Haplin [29]. We conducted genome-wide analyses of the fetal effect on fetal survival, with and without adjustment for parental interaction effects. An important note on terminology is that our dataset contains genotype data only for offspring who reached a gestational age well beyond the embryonic stage. Therefore, while our theoretical framework is centered around "embryo survival", our analyses focus on "fetal survival", and we use both terms accordingly. More details on data inclusion are provided later.

Similarly to how fetal effects may differ between non-ART and ART offspring, parental effects may also differ in non-ART compared to ART parents. Therefore, separate analyses were performed in two study samples of case-parent units: one with naturally conceived offspring and the other with offspring conceived through ART. We provide details on the model implementations, along with the results of the genome-wide analyses.

## 2 Materials and methods

### 2.1 Ethics statement

The study was conducted using data from the Norwegian Mother, Father, and Child Cohort Study (MoBa). The establishment of MoBa and initial data collection was based on a license from the Norwegian Data Protection Agency and approval from The Regional Committees for Medical and Health Research Ethics. The MoBa cohort is currently regulated by the Norwegian Health Registry Act. The current study was approved by The Regional Committees for Medical and Health Research Ethics (#2017/1362). The adult MoBa participants provided written statements of consent at inclusion, both for themselves and on behalf of their children [33]

### 2.2 Modeling strategies for reproductive trajectories

For a given couple, there may be multiple scenarios leading to a successful conception and having a baby, or trying without succeeding. To guide the statistical modeling, Fig 1 outlines a number of different reproduction trajectories that may lead to one of these two outcomes. The trajectory to reach a birth at a viable gestational age without the use of ART is marked with light gray boxes in Fig 1 and consists of the following three steps: 1) Each of the parents is independently biologically fertile; 2) as a couple, they are able to achieve fertilization without ART; and 3) the embryo survives, develops, and reaches maturity. Each of these steps can be disrupted by detrimental genetic or environmental factors. While environmental factors can, in principle, be adjusted for at each level, our modeling approach emphasizes how parental and fetal genes collectively contribute to the outcome.

Step 1 is the part most commonly associated with infertility, as numerous genetic factors are known to lead to female or male infertility. Couples where either or both partners are affected by genetically influenced infertility may decide to use ART. Consequently, if the ART procedure is successful, and the pregnancy reaches viability, the couple will end up in our second group, consisting of ART births at a viable gestational age. The trajectory toward a viable birth after ART use is marked with light purple boxes in Fig 1.

Step 2, however, requires a certain level of genetic compatibility between the parents. Notably, while each parent might be able to conceive with other partners, the combined genetic makeup of both parents may increase the difficulty of fertilization beyond what would be expected from their individual, genetically determined reproductive potential alone. Previously, our group explored various models of compatibility, or interactions, between parental alleles [24], inspired by an article that summarized all possible two-locus, two-allele penetrance models for dichotomous traits [34]. As an example, the "complementary" model assumes that opposite parental genotypes, i.e., a maternal *aa* combined with a paternal *AA*, or vice versa, can impact the likelihood of fertilization.

Step 3 primarily relates to embryo survival and early pregnancy losses. If a couple experiences frequent early pregnancy losses, they may decide to pursue ART treatment. An ART pregnancy could then be successfully established, progressing beyond the risk time window of early miscarriage, either because the ART procedure enhances early fetal survival or because the fertilized egg does not carry the deleterious allele combination responsible for previous early losses.

While Steps 1 and 2 above relate to the parental genotypes, Step 3 mainly focuses on the fetal genotype. However, parental and fetal effects may be aliased, meaning that parental effects could be mistaken for fetal effects, and vice versa. Examples of these scenarios are illustrated in Fig 2 for each of the three steps.

For Step 1, consider a scenario where maternal infertility is associated with an elevated frequency of a recessive allele *a* at a specific SNP. In ART triads, maternal *aa* genotypes would then be expected to occur more frequently than paternal *aa* genotypes within the same triads, and also more frequently than maternal *aa* genotypes in non-ART triads. Moreover, the distribution of the fetal genotypes would also be shifted accordingly in both ART and non-ART triads. For Step 1 illustrated in Fig 2, filled cells represent elevated maternal genotype frequencies in ART triads, resulting directly from genetic selection. Patterned cells indicate altered fetal genotype frequencies, which occur as an indirect consequence of this selection.

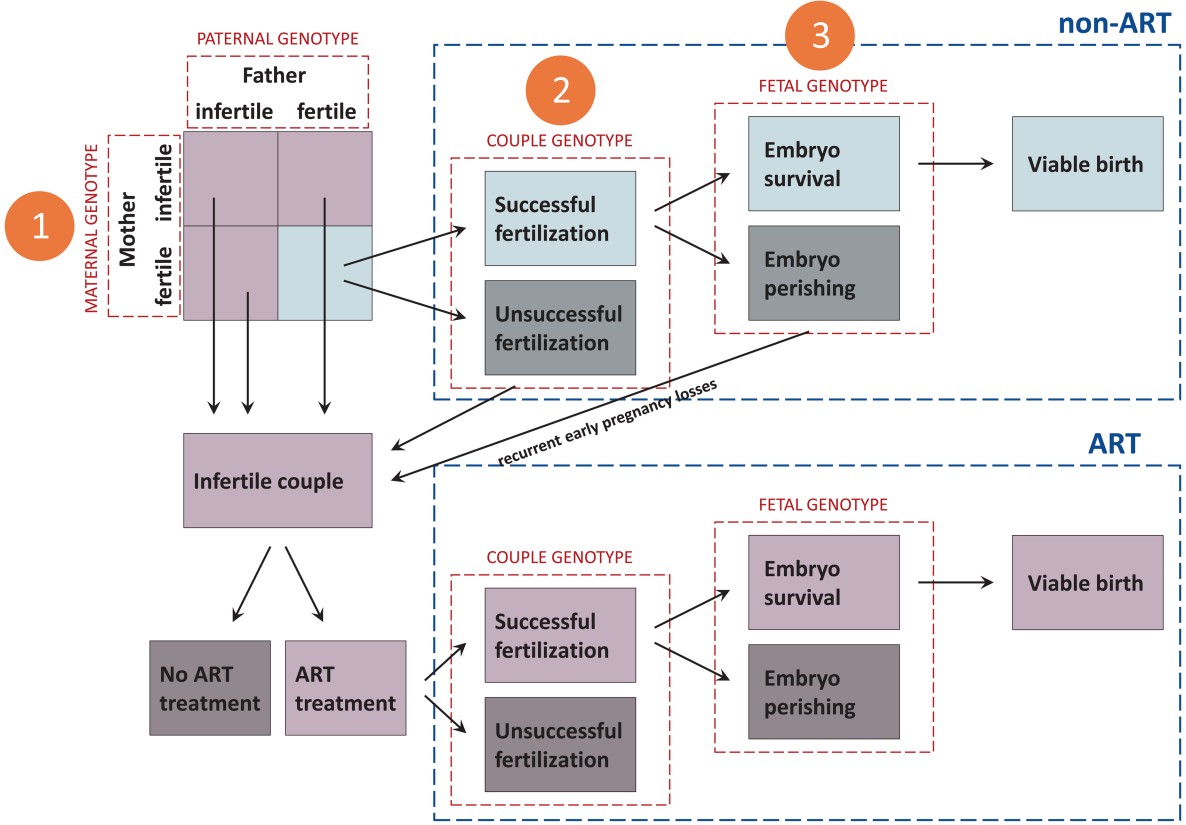

**Fig 1. Flowchart illustrating reproductive stages spanning the period before fertilization until birth and the genetic effects at play at each stage.** Each numbered stage is further illustrated with an example in Fig 2.

For Step 2, Fig 2 illustrates the selection under the complementary model, where opposite combinations of paternal alleles (filled cells of the same color) are more prevalent among ART triads than expected, and are subsequently less prevalent in non-ART triads. A maternal genotype of *AA* paired with a paternal genotype of *aa*, or vice versa, will always result in a heterozygous child. Consequently, this complementary interaction effect is indirectly reflected in the altered frequencies of heterozygous genotypes among children (patterned cells).

For Step 3, a deviation in the fetal allele distribution from what would be expected given the parental genotypes and Mendelian inheritance may suggest that the presence of a specific allele influences the probability of embryo survival. This phenomenon may occur in both ART and non-ART triads. However, couples experiencing frequent early losses due to the fetal genotypes being less compatible with early viability may proceed to ART treatment, and in such cases, the parents are more likely to carry the deleterious alleles themselves. ART triads may therefore exhibit distortions in both parental and fetal allele frequencies. For Step 3 in Fig 2, this scenario is exemplified by a fetal recessive genetic effect, which directly impacts the fetal *aa* frequencies (filled cells), whereas the corresponding parental allele frequencies are indirectly impacted by the selection of couples into the ART group (patterned cells).

## 2.3 Statistical models

From the above discussion, it is evident that ART triads reflect a mixture of two related study outcomes: 1) the risk of ART use at the parental level; and 2) survival at the embryo level. Previously, we modeled parental interactions in a parental

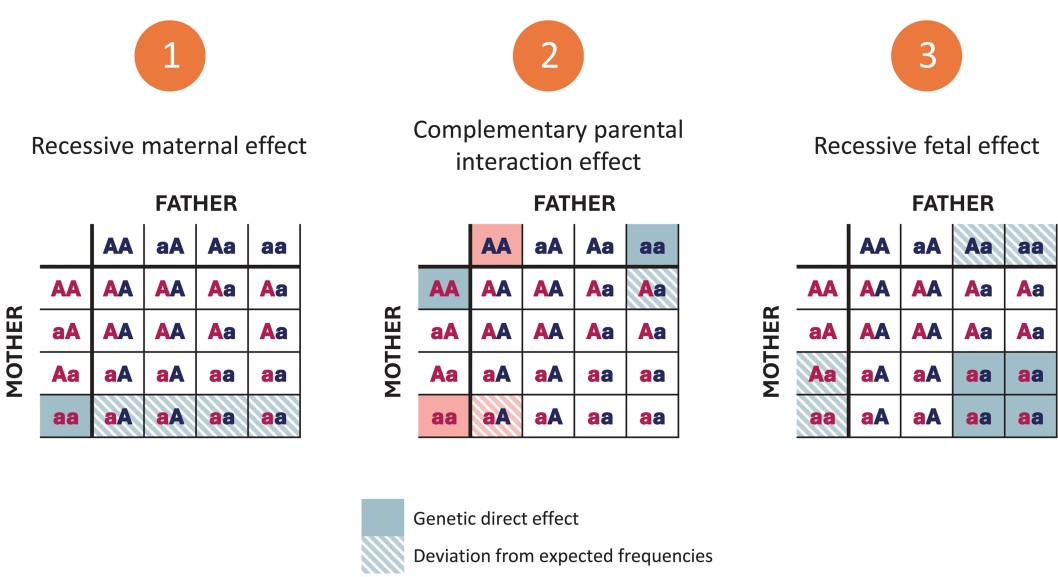

**Fig 2. Examples of genetic effects influencing the probability of fertilization and fetal survival, yielding a genetic selection into the ART group.** Solid cells represent the direct genetic effect, while patterned cells (diagonal lines) indicate genotypes whose frequencies may be altered by the direct genetic effects.

case-control setting, defining parents who used ART as cases and those who conceived naturally as controls [24]. We also demonstrated how our R package Haplin [29] can be used to analyze fetal survival in ART triads, estimating fetal effects as well as maternal and parent-of-origin effects, but without modeling the simultaneous effect of parental interactions [28]. In the current study, we extended the Haplin methodology to account for possible overlaps between parental and fetal genetic effects by jointly modeling parental interactions alongside fetal effects. By focusing the analyses on ART triads and non-ART triads separately, the risk of bias from population stratification is reduced, which allows for a more precise model. We will show that while within-triad analyses may still be exposed to biases from population substructure, such biases are less likely to have a serious impact on the results.

At a specific multi-allelic locus, assume that there are $H$ different alleles present in the population, and let $A_1, A_2, \ldots, A_H$ denote those alleles. Let $A_iA_j$ be the genotype of the mother in the triad, and $A_kA_l$ the genotype of the father. Here, $i, j, k, l \in \{1, 2, \ldots, H\}$, and for instance $i = j = 2$ would mean that the mother is homozygous with the $A_2A_2$ genotype. In the mother-father-child triad context we need to distinguish between those alleles that are transmitted to the child and those that are not. We write $M = (A_i, A_j)$ and $F = (A_k, A_l)$ for the *ordered* genotypes of the mother and father, respectively, where we assume that the first allele from each parent is not transmitted, whereas the second allele *is* transmitted to the offspring; hence, the offspring genotype can be expressed as $C = (A_j, A_l)$. To describe the population genotype distributions, we will for instance write $P(M) = P((A_i, A_j))$ for the proportion of mothers having the (ordered) $M$ genotype, i.e., as a simplified notation for the probability that a given mother has the ordered genotype $M$, and similarly $P(M, F)$ for the proportion of triads where the mother has the $M$ genotype and the father the $F$ genotype. The distribution of parental alleles in the background population, $P(M, F)$, is particularly important since it serves as a baseline for comparison when looking at the distribution actually observed in, for instance, the ART triads. For the modeling, we will consider the consequences of the following basic assumptions: 1) Mothers and fathers originate from populations with equal allele frequencies, with $p_1, p_2, \ldots, p_H$ denoting the (parental) background population frequencies of the $A_1, A_2, \ldots, A_H$ alleles, and thus $\sum_{h=1}^{H} p_h = 1$; 2) The parental population is in Hardy-Weinberg equilibrium (HWE), which implies that the unordered maternal genotype $A_iA_j$ has population frequency $2p_ip_j$ when $i \neq j$ and $p_i^2$ when $i = j$, and similarly for the paternal genotype; 3) There is

Mendelian transmission of alleles to the child, i.e., the two parental alleles have equal probability of transmission, implying that the two ordered genotypes $(A_i, A_j)$ and $(A_j, A_i)$ both have probability $p_i p_j$ when $i \neq j$; and 4) There is random mating between parents. This assumption finally implies that

$$P(M, F) = P(M) \cdot P(F) = p_i p_j p_k p_l, \tag{1}$$

thus describing the (ordered) genotype distribution in the parental population. Note that when written as ordered genotypes, we can write $P(M, F, C) = P(M, F)$ by notational convention, so that (1) actually describes the full expected distribution of triad genotypes in an unselected group. When analyzing ART triads, the validity of these assumptions can be checked using the non-ART triads. Below, we show below how deviations from these assumptions can be interpreted in terms of risk of ART use.

This formulation covers any multi-allelic locus with $H$ different alleles. Formulating the models for multi-allelic loci can be convenient, in particular to handle haplotypes constructed from a sequence of neighboring SNPs [29]. However, here it suffices to consider diallelic SNPs, with alleles $A_1$ and $A_2$ and where $p_1 + p_2 = 1$. The $A_1$ allele will be designated as the "reference allele" and $A_2$ as the "effect allele". In situations where we aim to estimate a relative risk parameter, we typically set $RR_1 = 1$ for the $A_1$ allele, and let $RR_2$ be the relative risk associated with the $A_2$ allele, a parameter to be estimated. For instance, at a C>T-SNP one might chose $A_1 = C$ as the reference allele and set $A_2 = T$ as the effect allele. In the current analyses, we typically define the major allele as the reference allele and the minor allele as the effect allele. We let $x_m$ and $x_f$ denote the parental allele dosages, so that $x_m \in \{0, 1, 2\}$ counts the number of $A_2$ alleles carried by the mother, and similarly with $x_f$ for the father and $x_c$ for the child.

To model and estimate the parental genotype distribution $P(M, F)$, we can write it using a log-linear formulation. A log-linear model assumes independent Poisson distributed counts of triads of each parental genotype combination, using a log-linear formulation for the expected numbers of each combination. As long as an intercept is included in the model, this is equivalent to writing it as a multinomial distribution, where the log probabilities follow a linear model:

$$\log(P(M, F)) = \alpha_0 + \alpha(x_m + x_f)$$
$$= \alpha_0 + \alpha x_m + \alpha x_f. \tag{2}$$

Here, $\alpha_0$ can be regarded as a scaling constant that ensures the probabilities sum up to 1. Also, $\alpha = \log(p_2/p_1)$ relates (2) to the multiplicative formulation in (1). Of particular note is that the linear dosage term $\alpha x_m$ signifies that maternal alleles are in HWE, and similarly for paternal alleles, although this restriction can easily be lifted. Furthermore, the common regression coefficient $\alpha$ for both $x_m$ and $x_f$ means that mothers and fathers derive from populations with the same allele frequencies, and the fact that there is no interaction term between the mother ($x_m$) and the father ($x_f$) equates to assuming random mating between the two.

If a given locus is involved in determining the risk of ART use for a couple, the parental genotype distribution $P(M, F|A)$ among ART couples will be shifted relative to the unconditional distribution. More specifically, using Bayes' formula, we have

$$P(M, F|A) = \frac{P(A|M, F)}{P(A)} \cdot P(M, F), \tag{3}$$

where the prevalence of ART, $P(A)$, can be considered a normalizing constant. We thus see that the probability of ART use, $P(A|M, F)$, determines how the conditional distribution $P(M, F|A)$ is shifted relative to $P(M, F)$.

In our previous work [24], we modeled the probability of ART use, $P(A|M, F)$, using a logistic regression model based on the genotypes of the individual parents, and combinations, i.e., interactions, of these. In the current setting, we use a

log-linear model for the risk of ART, parameterized as

$$\log(P(A|M,F)) = \beta_0 + \beta_m x_m + \beta_f x_f + \beta_{mf} x_m x_f, \tag{4}$$

where $x_m$ and $x_f$ are the dosage variables defined above, and $\beta_0$ is a scaling constant that makes the probabilities sum to 1. Note that the interaction effect $\beta_{mf}$ can be interpreted as an "incompatibility" between parental alleles, in that it impacts the probability of ART use above and beyond the individual contributions from each of the parents, as measured by $\beta_m$ and $\beta_f$, respectively.

The logistic model [24] used the non-ART triads as a baseline and compared the risk of ART use across different genotypes. However, in the current model, we can estimate parameters, in particular the interaction parameter $\beta_{mf}$, only based on the ART triads. This strategy is similar to an established approach, case-only analyses, to estimate gene-environment (GxE) interactions. In case-only analyses, under the assumption of independence of genes and environment in the population, GxE effects can be estimated [35,36]. In our setting we can achieve this by applying (2) and (4) to (3), thus deriving the log-linear formulation

$$\log(P(M,F|A)) = \beta_{00} + (\alpha + \beta_m)x_m + (\alpha + \beta_f)x_f + \beta_{mf} x_m x_f, \tag{5}$$

with $\beta_{00} = \alpha_0 + \beta_0 - \log(P(A))$. It should be noted that when we are looking at ART families alone, using (5), we cannot estimate $\beta_m$ and $\beta_f$ directly; only the difference between them can be estimated by calculating $(\alpha + \beta_m) - (\alpha + \beta_f) = \beta_m - \beta_f$. In other words, when studying ART triads, we cannot infer whether the increased risk of ART use is attributable the maternal genotype, the paternal genotype, or both. It is also clear that estimating the difference $\beta_m - \beta_f$ hinges on the assumption of a common $\alpha$ parameter for $x_m$ and $x_f$ from (2). Akin to a case-only study, the interaction effect $\beta_{mf}$ can be estimated based on the ART triads alone. However, note that this interaction effect may be aliased with non-random mating between parents. That is, if there is a correlation between parental genotypes, it would show up as a interaction term in (2).

The above model description essentially covers Steps 1 and 2 in Figs 1 and 2. Step 3, however, involves fetal genes, since alleles associated with early fetal survival would exhibit unexpectedly high frequencies in surviving children. Assuming that fetal survival during pregnancy depends both on fetal and maternal genes, the standard Haplin model [29] can be described as follows. Let $S$ denote the outcome that the fetus survives pregnancy, i.e., we now consider $S$ as the fetal "case" definition in a triad analysis. We then have for case-parent triads

$$P(M,F,C|S) = P(S|M,F,C)\,P(M,F,C) \cdot \frac{1}{P(S)}. \tag{6}$$

Here, $P(S)$ is again just a normalizing constant. The standard multiplicative model formulation for the penetrance used in Haplin is then

$$P(S|M,F,C) = B \cdot RR_j \cdot RR_l \cdot RR_i^{(M)} \cdot RR_j^{(M)}, \tag{7}$$

where $B$ is a baseline risk, $RR_j, RR_l$ are fetal relative risks associated with the fetal $A_j, A_l$ alleles, respectively, and $RR_i^{(M)}, RR_j^{(M)}$ are relative risks associated with the maternal alleles $A_i, A_j$, respectively. Since $A_1$ is assigned the role of reference allele, only $RR_2$ and $RR_2^{(M)}$ need to be estimated; $RR_1$ and $RR_1^{(M)}$ are both set to 1. More details about model implementation and interpretation, and the application of log-linear models to case-parent triads, are available in previous publications [28,29,37]. Note that if none of the alleles confers any risk, then $P(S|M,F,C)$ is constant, and

$$P(M,F,C|S) = P(M,F,C) = P(M,F),$$

so that the parental genotype distribution would be the same among couples that experience a successful birth as among all couples. Transforming (7) to log scale, we obtain

$$\log\left(P(S|M, F, C)\right) = \gamma_0 + \gamma_c x_c + \gamma_m x_m \tag{8}$$

for suitable $\gamma$-parameters, and applying (2) and (8) to (6) we obtain

$$\log(P(M, F, C|S)) = \gamma_{00} + \gamma_c x_c + (\alpha + \gamma_m)x_m + \alpha x_f,$$

with $\gamma_{00} = \alpha_0 + \gamma_0 - \log(P(S))$. Again, only the difference $(\alpha + \gamma_m) - \alpha = \gamma_m$ can be estimated.

We now combine the two model approaches into a joint model for ART parents with a surviving fetus. We again focus primarily on modeling the genotype distribution of the ART triads, this time conditioning on both $A$ and $S$, i.e., $P(M, F, C|S, A)$. As before, we can express this as

$$P(M, F, C|S, A) = P(S|M, F, C, A)\, P(A|M, F, C)\, P(M, F, C) \cdot \frac{1}{P(S, A)}.$$

Note that within ART triads with a surviving fetus, $P(S, A)$ is again just a normalizing constant in this expression. The penetrance model for fetal survival, $P(S|M, F, C, A)$, can be approached as in (8) above, except with the understanding that the risk parameters $\gamma_c$ and $\gamma_m$ are estimated conditional on ART use, i.e., it measures genetic effects on fetal survival in triads where ART is being used. The $P(A|M, F, C)$ and $P(M,F)$ parts can be modeled as described above. By combining (2), (4), and (8), we obtain the following log-linear model

$$\log(P(M, F, C|S, A)) = \delta + \gamma_c x_c + (\alpha + \beta_m + \gamma_m)x_m + (\alpha + \beta_f)x_f + \beta_{mf}x_m x_f, \tag{9}$$

with $\delta = \alpha_0 + \beta_0 + \gamma_0 - \log(P(S, A))$. To estimate model parameters, we again assume the observed count of each type of $(M,F,C)$ triad follows independent Poisson distributed variables, with expected values proportional to $P(M, F, C|S, A)$ in (9). As discussed above, to ensure identifiability of parameters, some parameters in (9) have to be restricted. While the difference $(\alpha + \beta_m + \gamma_m) - (\alpha + \beta_f) = \beta_m + \gamma_m - \beta_f$ can be estimated when again conditioning on the common parameter $\alpha$, the parameters $\beta_m$, $\gamma_m$, and $\beta_f$ cannot be distinguished. However, the two most important parameters in our analysis, $\gamma_c$ and $\beta_{mf}$, can be identified directly. These correspond to the two study outcomes described previously: the parental interaction parameter $\beta_{mf}$ captures the effect on the parental risk of requiring ART to conceive, while $\gamma_c$ reflects the effect of the fetal genotype on fetal survival.

The completed model (9) thus incorporates all three steps in Fig 1, and under suitable assumptions and parameter restrictions, the effects of the three steps can be separated. Violations of model assumptions may influence the effect estimates to varying degrees. The fetal effect is estimated conditional on parental genotypes and is therefore only modestly affected by population stratification [28]. In contrast, the parental interaction effect is more vulnerable to bias, particularly from population stratification and assortative mating. If two individuals in a couple are more genetically similar than expected by chance, this similarity may be misattributed by the model as a parental interaction effect. Furthermore, in relation to Step 3, it should be noted that while the model includes the fetal survival effect in ART triads, as expressed in (8), it does not explicitly model the indirect selection effect on parental alleles caused by possible previous pregnancy losses for the same parents; the $C$ in the formulas denotes the genotype of the successful ART birth, not any previously lost (and thus unobserved) siblings. It should also be recognized that although the model in (9) is derived within the context of only ART triads, comparable analyses can also be conducted within non-ART triads following similar reasoning.

Note that in all the above model descriptions, the dosage variables were added in a linear fashion, i.e., they are assumed to have a multiplicative dose-response effect. However, in many situations, it would make more sense to allow a free response model, i.e., a model that allows different relative risks for a single- and a double-dose of the effect allele. To achieve this, we can replace the dose variables by their components, for instance by replacing $x_c$ with $x_{c,1} + x_{c,2}$. Here, $x_{c,1}$ is 0 if the first allele of the child is $A_1$, and 1 if it is $A_2$, and $x_{c,2}$ similarly defined for the second allele of the child. Then, including a within-genotype allelic interaction in (8), the model $\gamma_c x_{c,1} + \gamma_c x_{c,2} + \gamma_{c,12} x_{c,1} x_{c,2}$ estimates the parameter $\gamma_{c,12}$ as a deviation from expectations of the multiplicative model; it thus allows for, for instance, recessive and dominant models to be fitted. Another example is if both $x_m$ and $x_f$ are split into two parts in (2); the corresponding model with within-genotype interactions allows deviations from the HWE assumption.

In line with our previous modeling of parental interaction effects [24], we implemented four different interaction models here: multiplicative, dominant, complementary, and threshold-based. The multiplicative interaction effect model was implemented as described in (9), based on the interaction model in (4). The dominant parental interaction effect model requires both parents to carry at least one copy of the minor allele, but the effect of one vs. two copies of the minor allele is assumed to be equal. To illustrate, we then define $\tilde{x}_m$ as 0,1,1 for the maternal genotypes $A_1 A_1, A_1 A_2, A_2 A_2$, respectively, and similarly for $\tilde{x}_f$. Thus, the interaction term $\tilde{x}_m \tilde{x}_f$ equals one in the case where both parents carry at least one copy of $A_2$, otherwise it is zero. These definitions of $\tilde{x}_m$ and $\tilde{x}_f$ would then be used in the interaction model (4) instead of $x_m$ and $x_f$, and (4) would again be inserted in (9), keeping $x_m$ and $x_f$ separate from the new $\tilde{x}_m$ and $\tilde{x}_f$. With a complementary parental interaction effect, we are targeting parents who are homozygous for the opposite alleles, i.e., at a specific locus, one parent is homozygous for the minor allele while the other is homozygous for the major allele. Finally, the threshold-based parental interaction effect assumes that the total number of minor alleles carried by both parents have an added effect on the outcome if they exceed a given threshold, in this setting equal to two. Similarly to the dominant model, both the complementary and the threshold-based models are implemented by modifying the interaction model in (4) accordingly before inserting it into (9). For more details on the parental interaction models and the specific coding of the complementary and threshold-based models, see Skodvin et al. [24].

Although not immediately evident from (9), the models assuming log-linear dose relationships, such as those containing the terms $\gamma_c x_c$ and $\beta_{mf} x_m x_f$ using the dosage variables, exhibit minimal aliasing between parental and fetal components, as explained in detail previously [28]. In our setting, fetal and parental effect estimates start impacting one another particularly for the complementary and threshold parental models in combination with a free model for the fetal genes. Consequently, these are the combinations we have focused on in our current analyses. The complementary and threshold models have some overlap but also differ in ways that are important to consider when interpreting the parental interaction effect estimates. Parents with the complementary genotype combination will also meet the threshold of two minor alleles as required by the threshold model. However, such couples likely represent only a minority among those identified by the threshold model. While the complementary model favors genetically dissimilar parents, the threshold model assigns larger interaction effects as the total number of minor alleles carried by both parents increases, regardless of whether these alleles come from similar or dissimilar genotypes.

We implemented the log-linear model described in (9) as an extension of the log-linear model already available in Haplin. By using maximum likelihood estimation and applying the expectation-maximization (EM) algorithm, missing data can be imputed, which means that Haplin also allows for analyzing incomplete triads, such as mother–child or father–child dyads. It also allows probability weighting of all possible configurations in triads with ambiguous transmission of alleles from parents to child, as well as imputation and weighting of possible haplotype configurations across multiple SNPs. For a detailed description of Haplin, see the original article [29].

## 2.4 Data and analytic framework

We used data from MoBa, a nationwide pregnancy cohort to which pregnant women were recruited from 1999 to 2008. Expectant mothers were invited to join the study at the time of their routine ultrasound examination, typically around week 17 of gestation, and the participation rate was 41% [38]. From 2001, fathers were also invited to the study. Currently, there are approximately 95,200 mothers, 75,200 fathers, and 114,500 children in the cohort. Blood samples were collected from the mothers and fathers at the time of the routine ultrasound examination and from the mothers and their newborn's umbilical cord at birth [39]. Due to the recruitment strategy of MoBa, the data are by default restricted to parents who were able to achieve a successful fertilization and carry the pregnancy to at least approximately week 17 of gestation. Hence, the use of ART serves as a key indicator of fertility challenges. Information on ART use was retrieved through data linkage with the Medical Birth Registry of Norway (MBRN) [40]. ART use was classified based on records indicating whether the pregnancy was achieved through in vitro fertilization or intracytoplasmic sperm injection. We included siblings in the analyses, but excluded multiple pregnancies. The two study samples eligible for analysis were as follows: for the non-ART sample, we had 42,047 mother-father-child triads; for the ART sample, we had 842 mother-father-child triads, 411 mother-child dyads, and 83 father-child dyads. Our data included a few instances of late stillbirths, with 18 occurring among non-ART pregnancies and fewer than 5 among ART pregnancies (exact number not provided to ensure privacy). All were reported to have occurred after gestational week 22.

Fig 1 shows that the group of ART parents comprises four subgroups from Step 1 based on parental fertility status: infertile mother, infertile father, both parents infertile, or neither parents infertile. Although additional information on infertility-related conditions is available in MBRN for some individuals, our data do not provide complete information to fully classify these subgroups. Nevertheless, our modeling strategy accounted for the main maternal and paternal effects, as demonstrated in (4). In other words, these effects are not ignored, even though they cannot be directly estimated within an ART triad design, as noted previously. Importantly, the maternal–paternal interaction effect—one primary focus of this study—can be estimated.

An extensive effort has been invested in genotyping as many individuals as possible in the entire MoBa cohort, with the process undertaken in multiple batches over the years [41]. Currently, the genetic data comprise approximately 77,000 mothers, 53,000 fathers, and 76,000 children. The genetic data have undergone thorough quality control as detailed by Corfield et al. [41]. We further processed the genetic data using PLINK v1.90 [42], applying a minor allele frequency (MAF) threshold of less than 1% and pruning SNPs in strong linkage disequilibrium ($r^2 > 0.95$), leaving approximately 2,300,000 SNPs for the current analysis. We used Haplin [30,31], implemented in R v4.0.4 [43], to analyze fetal effects on fetal survival, initially without adjusting for parental interaction effects. We then reanalyzed the fetal effects using the extension of the Haplin models including adjustments for parental interaction effects. The fetal effects were estimated by separate parameterizations of the single- and double-dose effects of the effect allele, with complementary and threshold-based parameterizations of the parental interaction effect. The analyses were conducted separately in the non-ART and ART samples. The effects were quantified as relative risk (RR) estimates, with 95% confidence intervals (CIs) and p-values. Further, we calculated the false discovery rate (FDR) q-values by applying the Benjamini-Hochberg method [44]. The Haplin framework also includes a test for Hardy-Weinberg equilibrium (HWE). All visualizations of the results were created using the R packages Haplin [30] and ggplot2 [45].

## 3 Results

Fig 3 shows the results of the analysis of single-dose fetal effects on fetal survival in the non-ART sample, without adjustment for parental interaction effects. Results for the corresponding double-dose fetal effect are presented in S1 Fig. The Manhattan plot in Fig 3A shows one distinct peak of neighboring SNPs on chromosome 6. The most significant SNP, rs114918746, had a p-value of $4.44 \times 10^{-16}$, three additional SNPs (rs113299093, rs2517588 and rs2905736) had p-values below the Bonferroni-corrected genome-wide significance threshold of $5 \times 10^{-8}$, and a fifth SNP (rs114011376)

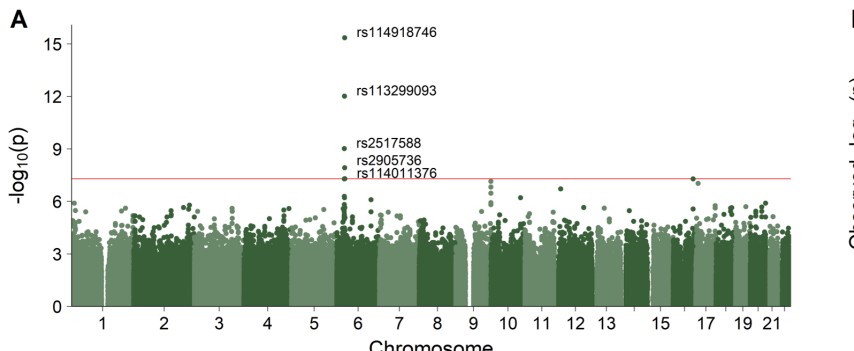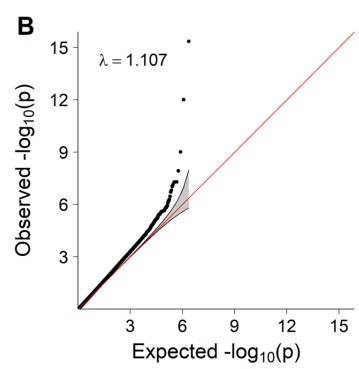

**Fig 3**. **The fetal effect estimated in the non-ART sample, unadjusted for parental interaction effects.** Panel **A** displays the negative $\log_{10}$-transformed $p$-values and the red line indicates the Bonferroni-corrected genome-wide significance threshold of $5 \times 10^{-8}$. Panel **B** shows the corresponding quantile-quantile (QQ) plot, along with the genomic inflation factor ($\lambda$).

had a $p$-value just slightly above the genome-wide threshold. Applying an FDR threshold of 0.1, we identified 10 additional significant SNPs, bringing the total to 15, including the five lead SNPs. The RR and MAF estimates for these SNPs are provided in Table 1. Most of the estimated MAFs for these SNPs were around 1-3%. However, a few more common variants were identified on chromosomes 12, 16, and 17, and their MAFs ranged from 5.6% to 11.5%. All RR estimates were below 1, ranging from 0.75 to 0.92. As indicated by the quantile-quantile (QQ) plot in Fig 3B, the $p$-value distribution was negligibly inflated ($\lambda$ = 1.107). Also included in Table 1 are the HWE $p$-values, and there was considerable variation in these $p$-values across the 15 significant SNPs, ranging from 0.006 to 0.993. The smaller $p$-values indicate violations of the HWE assumption. At a threshold of 0.05, three of the five lead SNPs (rs2517588, rs113299093, and rs114918746) deviated from the expected HWE distribution. The lead SNPs are located in or near genes such as Fibroblast Growth Factor Receptor 3 Pseudogene 1 (*FGFR3P1*), Ribosomal Protein L7 Pseudogene 4 (*RPL7P4*), Mediator of DNA Damage Checkpoint 1 (*MDC1*), HLA Complex Group 17 (*HCG17*), HLA Complex P5 (*HCP5*), MHC Class I Polypeptide-Related Sequence B (*MICB*), and Notch Receptor 4 (*NOTCH4*).

The same SNPs presented in Table 1 were re-analyzed in the non-ART sample, but this time adjusting for parental complementary and parental threshold-based interaction effects. The fetal effect RR estimates changed negligibly after these adjustments (top panel of Fig 4). The same SNPs were subsequently analyzed in the ART sample, first without adjustment for parental interaction effects and then with adjustment for parental complementary and parental threshold-based interaction effects (bottom panel of Fig 4). Most of the identified fetal effects were no longer statistically significant when analyzed in the ART sample. However, one SNP, rs113299093, showed a significant unadjusted fetal effect in the ART sample, similar to that observed in the non-ART sample. Its $p$-value of $1.09 \times 10^{-3}$ was below the candidate-SNP significance threshold of 0.05/15, based on the re-analysis of 15 SNPs. The RR estimates and $p$-values for the fetal and parental interaction effects of the SNPs listed in Table 1, analyzed in the models described above for both the non-ART and ART samples, are provided in S1 Table. A fetal effect RR estimate greater than 1 indicates an increased "risk" of fetal survival for carriers of the minor allele, whereas interpretation of parental interaction RRs depends on whether they are estimated within the ART or non-ART sample. In the ART sample, an RR above 1 corresponds to an increased risk of needing ART to conceive, i.e., a potentially reduced reproductive capacity, for couples carrying the interaction genotype combinations. Conversely, in the non-ART sample, an RR above 1 indicates an increased risk of *not* needing ART, reflecting enhanced reproductive capacity. Manhattan and QQ plots for the genome-wide analyses of parental interaction

**Table 1.** SNPs with a false discovery rate *q*-value below 0.1 from the analysis of fetal effects in the non-ART sample, unadjusted for parental interaction effects.

| SNP[1] | Type | Gene | Nearest gene(s) | Nearest gene distance (BP) | CHR | Minor allele | MAF | RR (95% CI) | p-value | FDR q-value[2] | HWE p-value[3] |
|---|---|---|---|---|---|---|---|---|---|---|---|
| rs2517588 | intron | HCG17 | | | 6 | a | 0.018 | 0.79 (0.73, 0.85) | 9.70e-10 | 7.53e-04 | 0.008 |
| rs61410672 | | | MICC | 7575 | 6 | a | 0.040 | 0.87 (0.82, 0.91) | 5.16e-08 | 1.73e-02 | 0.270 |
| rs145646034 | | | MICC | 15512 | 6 | a | 0.021 | 0.83 (0.78, 0.89) | 5.14e-07 | 9.21e-02 | 0.286 |
| rs113299093 | | | RPL7P4 MDC1 | 1076 1196 | 6 | t | 0.025 | 0.78 (0.73, 0.84) | 9.75e-13 | 1.14e-06 | 0.025 |
| rs148298273 | | | RNU6-1133P | 5857 | 6 | t | 0.024 | 0.85 (0.79, 0.90) | 6.07e-07 | 9.43e-02 | 0.949 |
| rs114918746 | | FGFR3P1 | | | 6 | t | 0.025 | 0.75 (0.70, 0.81) | 4.44e-16 | 1.03e-09 | 0.006 |
| rs2905736 | intron | | HCP5 MICB | 8007 9368 | 6 | a | 0.029 | 0.83 (0.78, 0.89) | 1.19e-08 | 6.93e-03 | 0.180 |
| rs114011376 | | | NOTCH4 | 3880 | 6 | a | 0.015 | 0.79 (0.72, 0.86) | 5.03e-08 | 1.73e-02 | 0.993 |
| rs72763276 | missense | TMEM210 | | | 9 | a | 0.016 | 0.81 (0.74, 0.88) | 3.42e-07 | 6.64e-02 | 0.209 |
| rs140430724 | intron | NDOR1 | | | 9 | t | 0.017 | 0.81 (0.75, 0.87) | 7.21e-08 | 2.10e-02 | 0.530 |
| rs35374928 | intron | EXD3 | | | 9 | a | 0.037 | 0.87 (0.82, 0.91) | 1.52e-07 | 3.55e-02 | 0.088 |
| rs4751986 | | | PNLIPRP3 RNU6-1090P | 29985 30241 | 10 | c | 0.023 | 0.84 (0.79, 0.90) | 6.08e-07 | 9.43e-02 | 0.012 |
| rs2300141 | intron | SLC2A3 | | | 12 | c | 0.115 | 0.92 (0.89, 0.95) | 1.91e-07 | 4.05e-02 | 0.622 |
| rs17677378 | intron | | MAF | 48677 | 16 | c | 0.090 | 0.90 (0.87, 0.94) | 5.21e-08 | 1.73e-02 | 0.963 |
| rs112874193 | intron | STX8 | | | 17 | a | 0.056 | 0.89 (0.85, 0.93) | 9.28e-08 | 2.40e-02 | 0.514 |

[1]Only one SNP from each pair in strong linkage disequilibrium ($r^2 > 0.95$) is presented in this table. [2]The *q*-values are estimated from the observed *p*-values using the Benjamini-Hochberg FDR method. [3]Hardy-Weinberg equilibrium test *p*-values.

effects in the non-ART and ART samples are presented in Figs 5 and 6. Corresponding plots for the single and double-dose fetal effects from the same models are presented in S2–S6 Figs. While not genome-wide significant, some noteworthy clusters of neighboring SNPs were apparent in these figures. Interestingly, these results also partially overlapped with those in our previous work [28]. More specifically, S5A and S6A Figs are comparable to Fig 2A in [28].

Although some caution is warranted when interpreting the validity and significance of the parental interaction effect estimates, we nevertheless opted to present these results here to motivate a methodological discussion on modeling strategies for such genetic effects. The analyses of parental complementary and parental threshold-based interaction effects in the non-ART sample identified 32 and 27 genome-wide significant SNPs, respectively; however, none of these SNPs overlapped. Details of these results, including MAF and RR estimates, as well as *p*-values, are provided in S2 Table. Most of the identified significant SNPs were rare, and had RR estimates mostly ranging from 1 to 3. In this context, an RR greater than 1 indicates that the specific parental genotype combination under study is more common than expected among parents who conceived without the use of ART. The significant SNPs associated with complementary interaction effects were broadly scattered across the genome, whereas some of the SNPs significantly associated with threshold-based interaction effects were more tightly clustered, particularly on chromosome 16. These latter SNPs are located in two genes, Zinc Finger CCHC-Type Containing 14 (*ZCCHC14*) and Junctophilin 3 (*JPH3*), neither of which have any previously published connections to infertility. Among the remaining significant parental interaction effects in the non-ART sample, two SNPs (rs35974259 and rs360673) are particularly noteworthy in that they are located in or near genes relevant to infertility. SNP rs35974259 on chromosome 13 is associated with a parental complementary interaction effect and is located near the gene for Sperm Acrosome Associated 7 (*SPACA7*). The protein product of this gene assists spermatozoa in breaking through the protective coating of cells surrounding the targeted oocyte during fertilization [46]. SNP

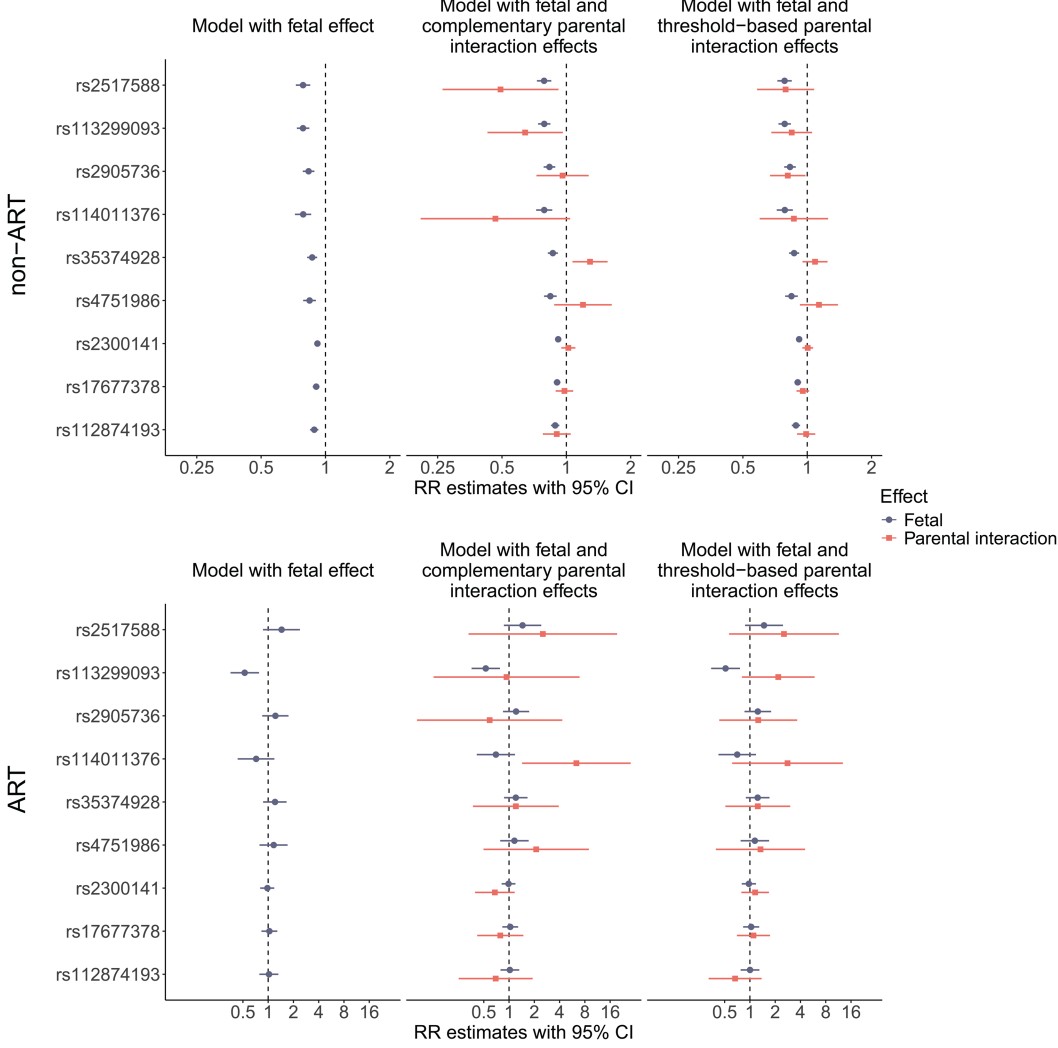

**Fig 4**. **Relative risks (RRs) for a subset of the SNPs presented in Table 1, estimated in the non-ART (top panel) and ART (bottom panel) samples, using a model with only the fetal effect (left) and models also including parental complementary (middle) and parental threshold-based (right) interaction effects.** For the six SNPs in Table 1 that were not included here, the parental interaction effect RRs in the ART sample could not be estimated properly, likely due to model convergence issues as a result of the small sample size.

rs360673 on chromosome 1 is associated with a parental threshold-based interaction effect and resides within Synaptonemal Complex Protein 1 (*SYCP1*), a gene known to play a crucial role in meiotic chromosome pairing and recombination [47].

As shown in the Manhattan and Miami plots in Figs 5 and 6, multiple SNPs were significantly associated with parental complementary and/or parental threshold-based interaction effects in both the non-ART and ART samples. There was no overlap between the two samples, as shown in the Miami plots directly comparing them in S8 Fig. The accompanying QQ plots revealed potential issues with the models; notably, the *p*-value distributions from the analyses in the non-ART sample were inflated, with genomic inflation factors of $\lambda = 1.383$ and $\lambda = 1.364$, respectively. On the other hand, for the results in the ART sample, while visual inspection of the QQ plots also indicated varying degrees of inflation, the corresponding $\lambda$-values were both less than 1 (0.697 and 0.828, respectively), suggesting that the overall distribution was *deflated*. This

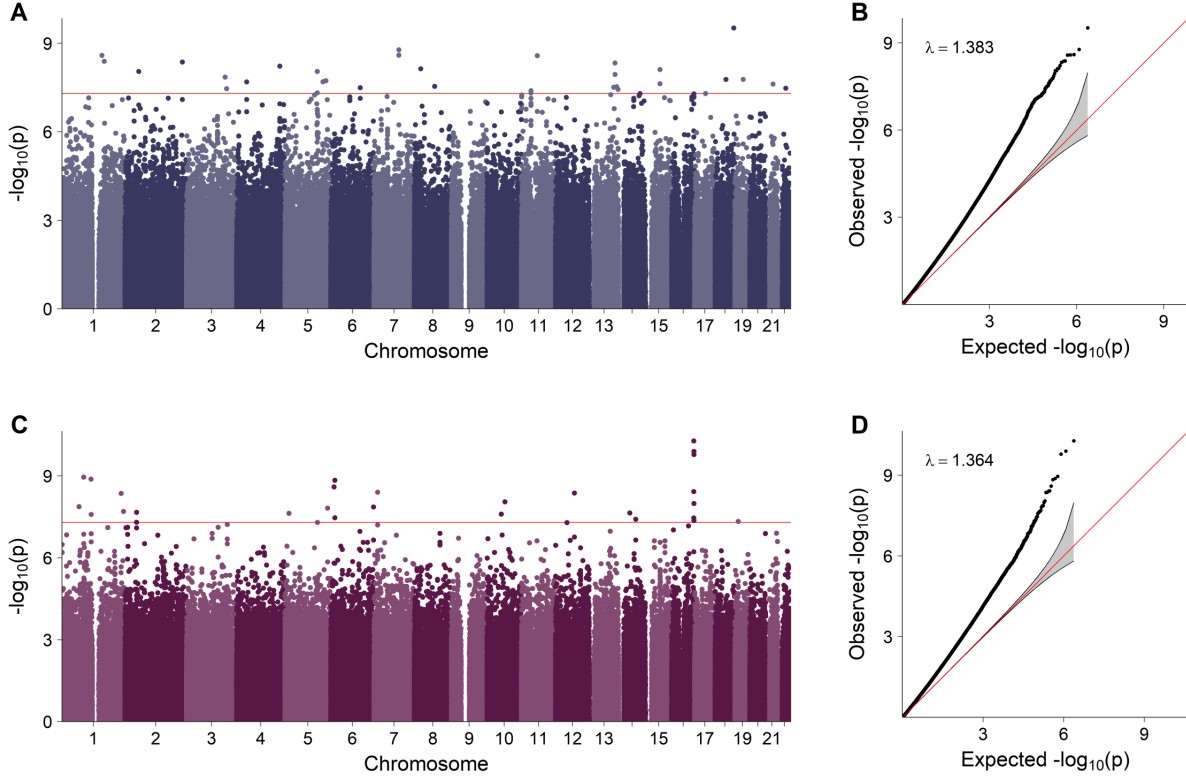

**Fig 5**. **Parental complementary (blue) and parental threshold-based (purple) interaction effects estimated in the non-ART sample.** Panels **A** and **C** display the negative $\log_{10}$-transformed *p*-values, and the red line indicates the Bonferroni-corrected genome-wide significance threshold of $5 \times 10^{-8}$. Panels **B** and **D** show the corresponding quantile-quantile (QQ) plots, along with the genomic inflation factor ($\lambda$).

was due to a disproportionately large number of *p*-values near 1, which was not evident from a visual inspection of the QQ plot. To explore these issues further, SNPs with a MAF below 5% were excluded and new QQ plots were generated (see S9 Fig). Excluding the rarest variants resulted in the inflation factors for the ART analyses being closer to 1 (1.093 and 1.099). Nevertheless, the results of the non-ART analyses remained substantially inflated, suggesting that the problem with the model may be due to factors other than low MAFs.

Within the ART sample, the results of the parental complementary and parental threshold-based interaction effects showed a greater degree of overlap, as illustrated by the Miami plot in Fig 6. Notably, one cluster of top SNPs on chromosome 7 was shared by both types of parental interaction effects. Further details, including RR estimates and *p*-values, for the SNPs with genome-wide significant parental interaction effects in the ART sample—either complementary, threshold-based, or both—are presented in S3 Table. Most of the RR estimates were quite large, which is likely due to low MAFs and small numbers of parental couples carrying the specific allele combinations. Nevertheless, RR estimates greater than 1 again indicate increased frequencies of the specific genotype combinations relative to those expected within this sample, i.e., they are more common among parents who used ART to conceive. Further investigation of these results revealed several relevant genes in the regions encompassing these SNPs, as shown in the regional association plots in Fig 7 and S7, including Actin Beta (*ACTB*), Fascin Actin-Bundling Protein 1 (*FSCN1*), and Ring Finger Protein 216 (*RNF216*).

We also conducted simulation-based power calculations for the model with adjustment for parental complementary interaction effects. Based on the framework of the hapSim function [48] in the Haplin R package, we simulated offspring-parent SNP data under scenarios with true parental complementary interaction effects ranging from RR = 1.2 to RR = 2.2,

PLOS Genetics

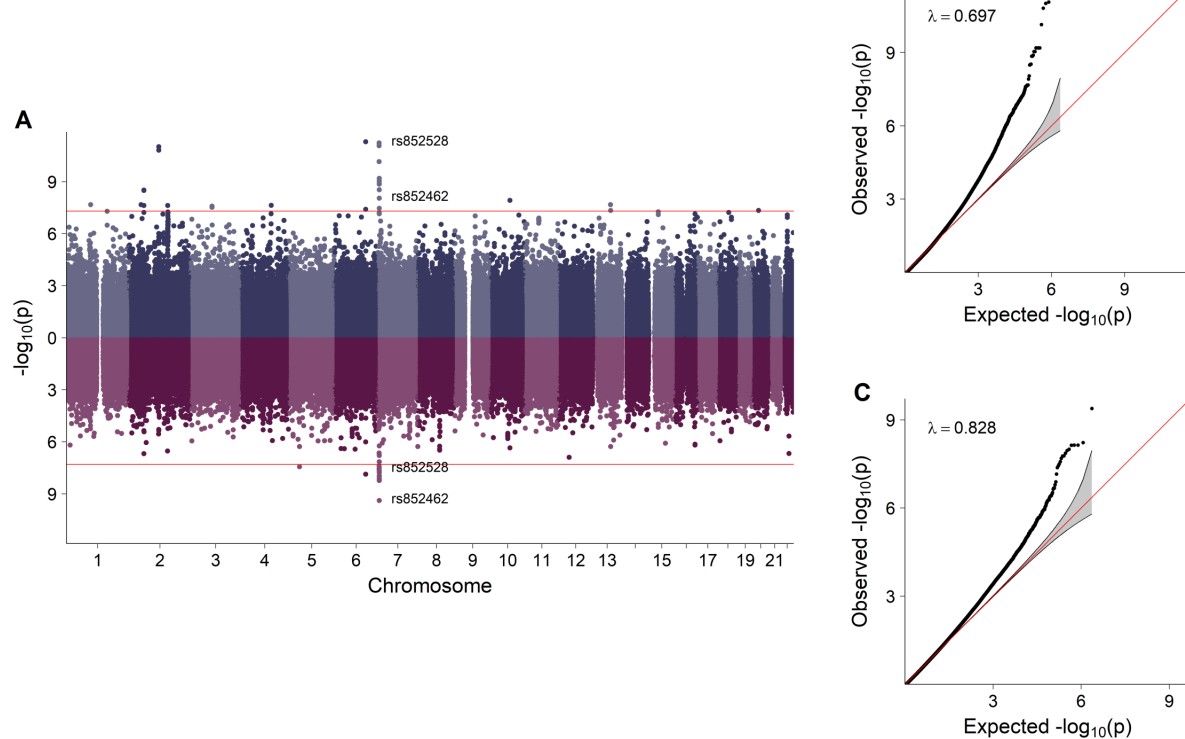

**Fig 6**. **Parental complementary (blue) and parental threshold-based (purple) interaction effects estimated in the ART sample.** Panel **A** shows the negative log$_{10}$-transformed *p*-values, with the red lines indicating the Bonferroni-corrected significance threshold of $5 \times 10^{-8}$. Panels **B** and **C** show the corresponding quantile-quantile plots, along with the genomic inflation factor ($\lambda$).

and with the fetal effect fixed at the null (RR = 1). We then analyzed the simulated data and estimated statistical power as the proportion of analyses in which a parental complementary interaction effect was significant at the Bonferroni-corrected threshold of $5 \times 10^{-8}$. Varying both MAF and sample size, we found that in a sample with a size comparable to our ART sample, the estimated power to detect a SNP with MAF = 10% and RR = 2.2 was 22%. If the MAF increased to 0.15, the estimated power increased to 73%. In a larger sample, with a size comparable to our non-ART sample, the estimated power to detect a SNP with MAF = 5% and RR = 1.5 was 79%. Results from these power calculations are presented in S10 Fig.

## 4 Discussion

In this study, we investigated parental and fetal genetic effects influencing infertility and fetal survival in family triads and dyads from the MoBa cohort. Case-parent units were categorized according to whether the offspring was conceived using ART or not, and separate analyses were conducted in these two groups. In the non-ART sample, we identified genome-wide significant fetal effects on fetal survival, primarily for several variants located on chromosome 6. These associations remained significant after adjustments for parental complementary and parental threshold-based interaction effects, confirming that these effects are fetal in origin. When re-analyzed in the ART sample, the results were either consistent with those in the non-ART sample or inconclusive. In other words, there was not enough evidence to conclude that the fetal effects observed in the non-ART offspring are absent in the ART offspring. One possible interpretation is that the observed fetal effect on fetal survival is not directly influenced by ART use. Many of the identified SNPs had low MAFs, which is as

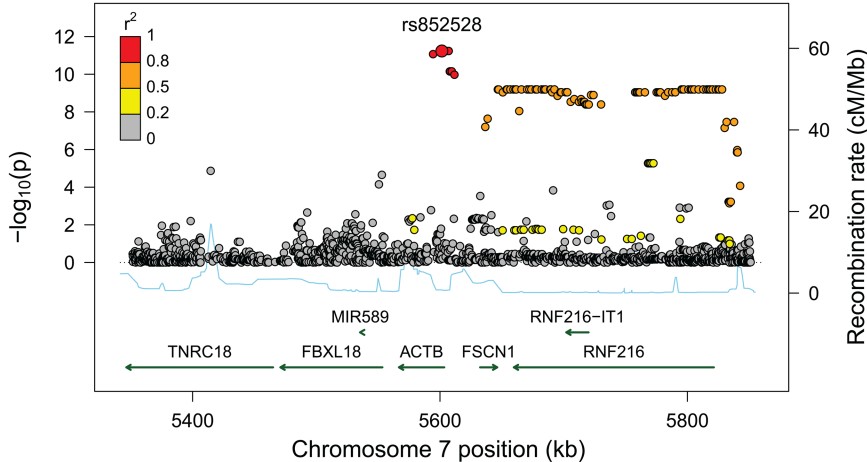

**Fig 7**. **Regional association plot spanning approximately 400 kb around the lead SNPs from the analysis of parental complementary interaction effects in the ART sample.** The green arrows indicate the direction of transcription for the genes in the region, while recombination rates are represented by blue lines. The color gradient reflects the degree of linkage disequilibrium, with associated $r^2$ values.

expected, as alleles with potentially detrimental effects on reproductive ability are unlikely to be common in the population. The RR estimates were all below 1, indicating that the effect (minor) alleles in the offspring were even less frequent than expected given the parental genotypes and Mendelian inheritance. The potential negative selection of these alleles suggests that they are associated with a lower probability of fetal survival. Furthermore, three of the SNPs with a genome-wide significant fetal effect deviated from HWE. A possible explanation for this is that the distribution of alleles associated with reproductive ability may shift over generations. Overall, due to the family-based study design, the fetal effect is inherently robust against population stratification, supporting the validity of these results and minimizing the risk of bias, for instance, from assortative mating.

Among the genetic variants identified as being associated with a fetal effect on fetal survival, several are located in or near genes involved in key reproductive processes, including fertility regulation, immune mechanisms, ovarian function, and early pregnancy development. For example, *MDC1* is linked to male fertility through its regulation of meiotic progression in spermatocytes [49,50]. *MICB* has been suggested to play a role in immune-related fertility mechanisms [51]. *HCP5*, a key long non-coding RNA (lncRNA), is implicated in premature ovarian insufficiency and regulates critical pathways involved in ovarian function and fertility [52]. *NOTCH4* is essential for several reproductive processes, including embryo implantation, placentation, and uterine function [53,54]. These genes are therefore relevant not only for the outcome under study—fetal survival—but also for other related phenotypes, such as the parents' ability to conceive. One possible explanation for the observed fetal effects in genes relevant for parental phenotypes is that alleles increasing the risk of infertility in parents may also, if passed down to the offspring, influence the probability of fetal survival. Furthermore, although the fetal effects were not attenuated after adjusting for parental complementary and parental threshold-based interaction effects, we cannot exclude the possibility that other types of parental effects could be involved.

We queried the 15 SNPs from Table 1 in the online database dbSNP [55] and found that several are classified as intronic variants, while others have no assigned functional consequence. Intronic variants may still affect gene expression by influencing mRNA stability, polyadenylation, miRNA binding, or other post-transcriptional regulatory processes. However, detailed functional and epigenetic annotations remain limited for many of these variants, and further investigation is needed to clarify their potential biological impact.

The parental interaction effects, as implemented in the models described here, are assumed to influence the overall ability to achieve fertilization, i.e., a study outcome that differs from fetal survival. When analyzed within the same

model, the aim is to separate fetal and parental effects. While the fetal effects capture the direct influence of the fetal genotype on fetal survival, the parental interaction effects, adjusted for fetal effects, reflect the couple's ability to achieve fertilization. For example, if a parental complementary interaction effect is identified with an RR estimate greater than 1 in the ART sample, the interpretation is that parents carrying the specified complementary genotype combination occur more frequently among ART triads than expected. In other words, such a result suggests an association between the parental complementary genotype combination and reduced reproductive capacity, assuming random mating. Since the parental interaction effect estimated in the non-ART sample reflects the opposite outcome—namely, enhanced reproductive capacity—one might expect the parental interaction RR estimates for the same SNPs to show opposite directions in the ART and non-ART samples. As shown in Fig 4 and S1 Table, this pattern occurred occasionally but was not consistent. Furthermore, it can be hypothesized that genetic variants detrimental to parental ability to achieve fertilization may also negatively impact the likelihood of fetal survival. If so, fetal and parental interaction effect estimates would be expected to align in direction within the non-ART sample, but diverge in the ART sample. However, this expectation was not consistently supported by the results presented in Fig 4 and S1 Table.

Compared with the fetal effect, parental interaction effect estimates are more susceptible to bias in case-triad analyses. Essentially, the four assumptions stated for the model definition in Sect 2.3 are not all necessarily met. The first assumption, that mothers and fathers originate from populations with equal allele frequencies, is not directly tested with our data. However, principal component analysis was performed on the full MoBa genetic dataset by Corfield et al., establishing that approximately 95% of MoBa participants are of European descent [41]. Nevertheless, MoBa participants represent a wide geographic range within Norway, and distinct geographic genetic subgroups have been identified in the Norwegian population [56]. In summary, we cannot claim that the first assumption is fully met, but any violations are likely limited in extent or effect.

The second model assumption concerns HWE, which we tested directly in our data. Because the HWE test pertains to the underlying data structure, it is only minimally influenced by which parameters are estimated. In other words, the HWE test p-values presented for the non-ART fetal effect results in Table 1 can also be used to assess the validity of the non-ART parental interaction effect estimates for the same SNPs. A few of the SNPs in Table 1 exhibited signs of departure from HWE, which could introduce bias into the corresponding parental interaction estimates.

Regarding the third model assumption of Mendelian transmission, it is important to recognize that deviations from the expected random segregation of alleles can occur at several stages, including fertilization or earlier. In our current analyses, such deviations may manifest as apparent genetic associations with early fetal viability. Wilcox et al. highlighted the assumption of Mendelian inheritance in case-parent triad studies, emphasizing its importance in accurately identifying genetic variants associated with disease in the offspring [57]. In the current study, we demonstrated that this assumption may not always hold and should therefore be specifically addressed in case-triad analyses.

The fourth model assumption listed in Sect 2.3, relates to random mating. This condition is commonly hard to fulfill in genetic association studies such as ours, and previous research has indeed demonstrated assortative mating in several traits for couples in the MoBa cohort in particular [58]. While such violations are less problematic for the estimation of fetal effects, they may introduce bias in the parental interaction effect estimates. Moreover, the genetic selection of parents into the non-ART or ART groups may depend on factors unrelated to infertility, potentially biasing the results. If specific genotype combinations accumulate in couples as a result of positive or negative assortative mating, population stratification, or selection bias, they may manifest as false-positive parental interaction effects. This is likely a greater issue for SNPs with low MAFs, as the interaction depends on two loci rather than one, and genotype frequencies for these SNPs have the potential to deviate more relative to the expected frequencies under Mendelian inheritance. Nevertheless, variants that influence parental ability to conceive, i.e., their fecundity, are likely to be rare in the population. The analyses of parental complementary and parental threshold-based interaction effects primarily identified lead SNPs with low MAFs in both the large non-ART sample and the substantially smaller ART sample.

The QQ plots for the parental interaction effects in the non-ART sample revealed model-related issues that could not be resolved by excluding low-frequency variants. Interestingly, the impact of excluding rare variants differed between the non-ART and ART samples, with QQ plots showing improved model calibration after excluding SNPs with low MAF in the ART sample only. Since this effect was not observed in the non-ART sample, it could simply have been due to chance. A possible explanation could be that the impact on genomic inflation caused by confounding factors could be larger in a larger sample size. Analyses of parental interaction effects in the ART sample are likely subject to similar overall issues as those in the non-ART sample. Furthermore, certain subtypes of infertility may be polygenic, which could contribute to the genomic inflation observed in the QQ plots. Polygenicity could be assessed using methods such as LD score regression [59]. However, when the unit of analysis is parental couples, as in this study, no immediately applicable LD score regression approach is available.

Although the interpretation of the parental interaction analyses is not straightforward, the choice of parameterizations still merits further discussion. The complementary and threshold-based parameterizations presented here are just two examples of how parental incompatibility may manifest as a genetic effect affecting reproductive ability. Several other parameterizations could also be used to model relevant parental interaction effects, some of which were explored in case-control analyses in our previous work [24]. There, we showed that different parameterizations can yield distinct results, underscoring the value of a broad analytical approach incorporating multiple models. The choice of model may be guided by prior hypotheses about the underlying biological mechanisms. For instance, if a parental genetic incompatibility is suspected to influence reproductive capacity, the complementary model may be particularly appropriate. Overall, the complementary and threshold-based parameterizations of the parental interaction effect exhibit some overlap. Parents who harbor the complementary interaction genotype combinations—where one parent is homozygous for the effect allele and the other for the reference allele—also meet the criteria for the threshold-based interaction combinations, where the combined number of effect alleles in both parents is required to be 2 or higher. In other words, a true complementary effect could be detected using a threshold-based parameterization, and vice versa. This may not be true for other threshold values. For example, a threshold of 3 excludes the complementary genotype combinations, while a threshold of 1 includes all parental genotype combinations except the combination where both parents are homozygous for the reference allele. A promising direction for future research could be to investigate when and why the complementary and threshold-based models yield overlapping results—and when they do not—as this could help inform model choice in other settings. It is also important to acknowledge that additional genetic mechanisms not captured by our models may influence the outcomes. In the context of infertility and fetal survival, a maternal-fetal genotype incompatibility, as described by Sinsheimer et al. may also be relevant [60]. Models accounting for maternal-fetal interactions, as well as other types of genetic effects, have been summarized by Ainsworth et al. [61]. Typically, attempts to estimate multiple genetic parameters are constrained by the available degrees of freedom.

Overlapping results between the two parental interaction models were indeed observed in the ART sample, where several of the same significant SNPs were identified as having both parental complementary and parental threshold-based interaction effects. Of note, these SNPs are located in a region containing several genes associated with infertility. For instance, *ACTB* is involved in proper sperm formation and male fertility through its role in organizing the mitochondrial sheath during spermiogenesis [62,63]. *FSCN1*, an actin filament-bundling protein, is also essential for male fertility and spermatogenesis [64,65]. Similarly, *RNF216* encodes an E3 ubiquitin ligase that plays a crucial role in male fertility through regulating spermatogenesis, meiosis, and PKA stability in the testes [66,67]. The phenotypic relevance of these genes supports the notion that these associations are true rather than spurious. For the non-ART analyses, no significant SNPs overlapped between the complementary and threshold-based models. One possible explanation for why an overlap in results for the parental interaction effect was observed only in the ART sample is that the parental phenotype studied in this group—risk of needing to use ART—is more specific and well-defined than the contrasting phenotype examined in the non-ART sample. Another possibility is that if some of the findings are influenced by confounding factors, such as assortative mating, these factors may be more pronounced in the ART group.

The high quality of the genetic and registry data combined with the large sample size, particularly for the non-ART sample, are among key strengths of this study. There are likely only a few other datasets in the world where a genome-wide case-parent triad analysis at this scale is feasible. In contrast, although the ART sample also ranks among the largest such datasets globally, the sample size remains relatively modest in a genome-wide context, limiting the statistical power to detect small or moderate effects in this group. The substantial difference in sample size between the ART and non-ART samples limits the reliability of comparisons between these groups. Furthermore, the relatively low initial participation rate in MoBa suggests that the cohort may not be fully representative of the general population. For instance, MoBa participants generally have higher levels of education compared to the broader population [68]. However, Nilsen et al. analyzed MoBa data and demonstrated that bias in prevalence estimates does not necessarily imply bias in the estimated associations between exposures and outcomes [69]. During the years of recruitment into the MoBa cohort (1999–2008), the overall Norwegian population grew [8], and ART use increased modestly both in the population [8] and within the cohort (See S11 Fig). However, the impact on our analyses is likely minimal given the relatively homogeneous ancestry of the cohort. As is often the case, genotypes from pregnancy losses were unavailable and therefore could not serve as a suitable control group for these analyses. Given this limitation, the case-parent triad design utilized here represents one of the most robust approaches available. Potential analyses to follow up on or complement our work include studies using other measures of reproductive ability, such as time to pregnancy, and functional analyses of the genetic associations reported in this study.

In conclusion, we identified fetal effects on fetal survival that persisted after adjusting for parental interaction effects in the non-ART sample. The evidence was inconclusive regarding whether these effects were also present in the ART sample. Nevertheless, the reliability of the observed fetal effects in the non-ART offspring is supported by the sample size, the robustness of the fetal effect estimates due to the case-triad study design, and the relevant genetic context. We also observed parental interaction effects in previously reported infertility-related genes, especially in the ART sample. As with most observational studies, our findings may have been influenced by biases that are difficult to account for in the models. This study is among the first to examine the joint estimation of parental interaction effects and fetal effects using a family-based study design. The identification of numerous genes related to reproductive ability supports the validity of the approach. Future studies are nevertheless needed for replication and validation of our findings.

## Supporting information

**S1 Table. The SNPs presented in Table 1 re-analyzed in the non-ART and ART samples, unadjusted for parental interaction effects and adjusted for parental complementary and parental threshold-based interaction effects.** (XLSX)

**S2 Table. SNPs with genome-wide significant parental interaction effects in the non-ART sample.** (XLSX)

**S3 Table. SNPs with genome-wide significant parental complementary and/or parental threshold-based interaction effects in the ART sample.** (XLSX)

**S1 Fig. The fetal double-dose effect estimated in the non-ART sample, unadjusted for parental interaction effects.** Panel **A** displays the negative $\log_{10}$-transformed $p$-values, and the red line indicates the Bonferroni-corrected genome-wide significance threshold of $5 \times 10^{-8}$. Panel **B** shows the corresponding quantile-quantile (QQ) plot, along with the genomic inflation factor ($\lambda$). (TIF)

**S2 Fig. The fetal single-dose (panels A and B) and double-dose (panels C and D) effect estimated in the non-ART sample, adjusted for parental complementary interaction effects.** Panels **A** and **C** display the negative

log$_{10}$-transformed $p$-values, and the red line indicates the Bonferroni-corrected genome-wide significance threshold of $5 \times 10^{-8}$. Panels **B** and **D** show the corresponding quantile-quantile (QQ) plots, along with the genomic inflation factor ($\lambda$).
(TIF)

**S3 Fig. The fetal single-dose (panels A and B) and double-dose (panels C and D) effect estimated in the non-ART sample, adjusted for parental threshold-based interaction effects.** Panels **A** and **C** display the negative log$_{10}$-transformed $p$-values, and the red line indicates the Bonferroni-corrected genome-wide significance threshold of $5 \times 10^{-8}$. Panels **B** and **D** show the corresponding quantile-quantile (QQ) plots, along with the genomic inflation factor ($\lambda$).
(TIF)

**S4 Fig. The fetal single-dose (panels A and B) and double-dose (panels C and D) effect estimated in the ART sample, unadjusted for parental interaction effects.** Panels **A** and **C** display the negative log$_{10}$-transformed $p$-values, and the red line indicates the Bonferroni-corrected genome-wide significance threshold of $5 \times 10^{-8}$. Panels **B** and **D** show the corresponding quantile-quantile (QQ) plots, along with the genomic inflation factor ($\lambda$).
(TIF)

**S5 Fig. The fetal single-dose (panels A and B) and double-dose (panels C and D) effect estimated in the ART sample, adjusted for parental complementary interaction effects.** Panels **A** and **C** display the negative log$_{10}$-transformed $p$-values, and the red line indicates the Bonferroni-corrected genome-wide significance threshold of $5 \times 10^{-8}$. Panels **B** and **D** show the corresponding quantile-quantile (QQ) plots, along with the genomic inflation factor ($\lambda$).
(TIF)

**S6 Fig. The fetal single-dose (panels A and B) and double-dose (panels C and D) effect estimated in the ART sample, adjusted for parental threshold-based interaction effects.** Panels **A** and **C** display the negative log$_{10}$-transformed $p$-values, and the red line indicates the Bonferroni-corrected genome-wide significance threshold of $5 \times 10^{-8}$. Panels **B** and **D** show the corresponding quantile-quantile (QQ) plots, along with the genomic inflation factor ($\lambda$).
(TIF)

**S7 Fig. Regional association plot spanning approximately 400 kb around the lead SNPs from the analysis of parental threshold-based interaction effects in the ART sample.** The green arrows indicate the direction of transcription for the genes in the region, while recombination rates are represented by blue lines. The color gradient reflects the degree of linkage disequilibrium, with associated $r^2$ values.
(EPS)

**S8 Fig. Reorganization of Figs 5 and 6 for illustrative purposes.** The parental interaction effect are shown for the complementary model (panel **A**) and the threshold-based model (panel **B**) in the ART sample (blue) and the non-ART sample (red). The negative log$_{10}$-transformed $p$-values are displayed, and the red lines indicate the Bonferroni-corrected genome-wide significance threshold of $5 \times 10^{-8}$.
(TIF)

**S9 Fig. Quantile-quantile (QQ) plots for the parental interaction effect estimated in the ART (panels A and B) and non-ART (panels C and D) samples.** Panels **A** and **C** display the parental complementary interaction effects, while panels **B** and **D** display the parental threshold-based interaction effects. The shaded area represents a 95% confidence interval band for the null hypothesis $p$-value distribution. SNPs with MAF lower than 5% are excluded.
(TIF)

**S10 Fig. Simulation-based power calculations for the parental complementary interaction effect, based on 1,500 simulations of family-triad SNP data for each combination of sample size, MAF, and true parental complementary**

**interaction relative risk (RR).** Statistical power was estimated as the proportion of analyses in which the parental complementary interaction effect was significant at the Bonferroni-corrected threshold of $5 \times 10^{-8}$.
(EPS)

**S11 Fig. Annual absolute frequency and proportion of newborns conceived via ART in MoBa.**
(EPS)

## Acknowledgments

The Norwegian Mother, Father, and Child Cohort Study is supported by the Norwegian Ministry of Health and Care Services and the Ministry of Education and Research. We are grateful to all the participating families in Norway who take part in this on-going cohort study. We thank the Norwegian Institute of Public Health (NIPH) for generating high-quality genomic data. This research is part of the HARVEST collaboration, supported by the Research Council of Norway (#229624). We also thank the NORMENT Centre for providing genotype data, funded by the Research Council of Norway (#223273), South East Norway Health Authorities and Stiftelsen Kristian Gerhard Jebsen. We further thank the Center for Diabetes Research, the University of Bergen for providing genotype data and performing quality control and imputation of the data funded by the ERC AdG project SELECTionPREDISPOSED, Stiftelsen Kristian Gerhard Jebsen, Trond Mohn Foundation, the Research Council of Norway, the Novo Nordisk Foundation, the University of Bergen, and the Western Norway Health Authorities. All analyses were performed using digital laboratories in HUNT Cloud at the Norwegian University of Science and Technology, Trondheim, Norway. We are grateful for outstanding support from the HUNT Cloud community. AI-assisted language editing using ChatGPT (OpenAI) was employed to improve the clarity and grammar of the manuscript. The authors take full responsibility for the content and interpretation.

## Author contributions

**Conceptualization:** Siri N. Skodvin, Miriam Gjerdevik, Julia Romanowska, Astanand Jugessur, Håkon K. Gjessing.

**Data curation:** Alexandra Havdahl.

**Formal analysis:** Siri N. Skodvin, Håkon K. Gjessing.

**Funding acquisition:** Siri E. Håberg.

**Investigation:** Siri N. Skodvin, Miriam Gjerdevik.

**Methodology:** Siri N. Skodvin, Miriam Gjerdevik, Rolv Terje Lie, Håkon K. Gjessing.

**Project administration:** Siri E. Håberg, Håkon K. Gjessing.

**Resources:** Alexandra Havdahl, Håkon K. Gjessing.

**Software:** Rolv Terje Lie, Håkon K. Gjessing.

**Supervision:** Miriam Gjerdevik, Julia Romanowska, Astanand Jugessur, Håkon K. Gjessing.

**Validation:** Siri N. Skodvin, Astanand Jugessur.

**Visualization:** Siri N. Skodvin.

**Writing – original draft:** Siri N. Skodvin, Astanand Jugessur, Håkon K. Gjessing.

**Writing – review & editing:** Siri N. Skodvin, Miriam Gjerdevik, Julia Romanowska, Siri E. Håberg, Rolv Terje Lie, Astanand Jugessur, Håkon K. Gjessing.

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
