## [Decision Letter · Decision Letter 0]

13 Jun 2025

PGENETICS-D-25-00416

Statistical methods to disentangle genetic effects influencing infertility and early fetal viability with a genome-wide application

PLOS Genetics

Dear Dr. Skodvin,

Thank you for submitting your manuscript to PLOS Genetics. After careful consideration, we feel that it has merit but does not fully meet PLOS Genetics's publication criteria as it currently stands. Therefore, we invite you to submit a revised version of the manuscript that addresses the points raised during the review process. 

Please submit your revised manuscript within 60 days Aug 12 2025 11:59PM. If you will need more time than this to complete your revisions, please reply to this message or contact the journal office at plosgenetics@plos.org. Please include the following items when submitting your revised manuscript:

We look forward to receiving your revised manuscript.

Kind regards,

Yun Li

Academic Editor

PLOS Genetics

Gregory Cooper

Section Editor

PLOS Genetics

Aimée Dudley

Editor-in-Chief

PLOS Genetics

Anne Goriely

Editor-in-Chief

PLOS Genetics

**Journal Requirements:**

4) Please ensure that the funders and grant numbers match between the Financial Disclosure field and the Funding Information tab in your submission form. Note that the funders must be provided in the same order in both places as well.

**Reviewers' comments:**

Reviewer's Responses to Questions

**Comments to the Authors:**

Reviewer #1: This manuscript presents a statistical framework to disentangle genetic effects of parents from the fetus on fertilization and early fetal survival. By leveraging data from both ART and non-ART families, the authors identified fetal genetic factors and parental interactions that might be linked to early viability. A key challenge the authors tried to address is the potential aliasing of parental and fetal effects. The statistical model is clearly described with a well-defined set of estimable parameters and assumptions. The authors applied several modeling strategies and compared models with and without parental interaction for the ART and non-ART cohorts. Overall, I think the methodology is well presented. I have a few comments and questions about the interpretation and results that can be improved before considering publication.

Comments

1. It will be helpful to explain more explicitly how the proposed method extends or differs from the Haplin framework for the readers to understand the novelty of the proposed approach.

2. Can the authors give explicit interpretation of the parental interaction effect estimates? There is some discussion in the Discussion section, but it will be more helpful to have this earlier in the manuscript. What caveats should be kept in mind? Additionally, how do different interaction modeling strategies change the interpretation?

3. In Figure 4, modeling both fetal and parental interaction effects results in sometimes same and sometimes opposite directions of relative risk. Does this have biological or statistical implications? More guidance on how to interpret such results would be helpful.

4. For the identified SNPs, besides which gene regions they are located in or close to, are they located in intronic, exonic, or regulatory regions? What about variant effect type (e.g., missence, nonsence, etc.)? More functional and epigenetic annotations would aid the biological interpretation.

5. The genomic inflation factors observed in the study are briefly addressed in the Discussion, but more elaboration is needed on the potential causes of inflation or deflation, as well as on the robustness of the findings. Have approaches such as LD score regression been considered to distinguish polygenicity vs. modeling or confounding issues? For the ART cohort, it’s likely the sample size is too small for it to have sufficient power, which is a limitation of the study that should be brought up.

6. Have the authors checked the underlying assumptions in the model with the data?

7. Related to comment 2, in the Discussion section the manuscript talks about interpretations of parental interaction effects and mentions “assuming non-random mating” (line 475), which appears to contradict earlier statements and model assumptions. Clarification is needed regarding the modeling assumptions and interpretation of parameters.

8. Given the multiple interaction modeling approaches (complementary, threshold-based, etc.), in practice, how should one select among them? With many unmeasured confounders and uncontrolled factors, the manuscript would benefit from a clearer discussion of model validity, usability, and interpretability in real use cases.

Reviewer #2: Review uploaded as an attachement

Reviewer #3: Thank you for the opportunity to review this manuscript. The authors analyzed the ART and non-ART sample and compared the results between two groups, between with and without adjustment for interaction effects, and among different choices of interaction effects. Overall, the manuscript is well written and provides valuable insights. However, some issues need to be addressed before publication.

1. The model structure involves P(A|M,F,C), while the model is fitted separately for A=0 and A=1. As the interaction effects are used to estimate P(A|M,F,C), I am not confident in the reliability of the coefficient estimate for the interaction effects, when fitting separately for A=0 and A=1. In other words, since the model indicates that beta_mf is the coefficient for the effect of the interaction term on the risk of ART, it is unreasonable to estimate the coefficient only using the ART or non-ART samples.

2. In all the formulas derived in Section 4.1, authors used the multiplicative interaction effect while in real data analysis, they instead used complementary and threshold-based parental interaction effects. It is suggested to use complementary and threshold-based parental interaction effects in the formulas derived in Section 4.1 or include more details, such as the formulas for these two interaction effects.

3. On page 5, Paragraph 2, the notations are not clear to me. It seems that A_h denotes different alleles (rather than a variable), but A_i, A_j, …, appears to be variables.

4. Considering there are only 842 samples in the ART sample, do you worry about the reliability of your analysis in the ART sample? In addition, the significant discrepancy between the sample sizes in the two groups may affect the comparison between results from the ART and non-ART sample. More discussion on it is needed.

5. Do you have any explanation for the lack of overlap between the SNPs selected by the complementary and threshold-based parental interaction effects? For example, if the threshold relaxed, is there any overlap?

6. Before Table 1, “[htbp]” should be removed.

**Have all data underlying the figures and results presented in the manuscript been provided?**

Reviewer #1: None

Reviewer #2: Yes

Reviewer #3: None

PLOS authors have the option to publish the peer review history of their article (what does this mean?). If published, this will include your full peer review and any attached files.

Reviewer #1: No

Reviewer #2: No

Reviewer #3: No

**Figure resubmission:**
---

## [Decision Letter · Decision Letter 1]

10 Nov 2025

Dear Dr Skodvin,

We are pleased to inform you that your manuscript entitled "Statistical methods to disentangle genetic effects influencing infertility and early fetal viability with a genome-wide application" has been editorially accepted for publication in PLOS Genetics. Congratulations!

Yours sincerely,

Yun Li

Academic Editor

PLOS Genetics

Gregory Cooper

Section Editor

PLOS Genetics

Aimée Dudley

Editor-in-Chief

PLOS Genetics

Anne Goriely

Editor-in-Chief

PLOS Genetics

BlueSky: @plos.bsky.social

Comments from the reviewers (if applicable):

Reviewer's Responses to Questions

**Comments to the Authors:**

Reviewer #1: Thank the authors for addressing my comments. I have one minor comment:

Lines 571–574: In the discussion of LD score regression, please cite the correct reference (DOI: 10.1038/ng.3211).

Reviewer #2: Thank you for addressing my comments.

Reviewer #3: The authors have made necessary modifications.

**Have all data underlying the figures and results presented in the manuscript been provided?**

Reviewer #1: Yes

Reviewer #2: Yes

Reviewer #3: None

PLOS authors have the option to publish the peer review history of their article (what does this mean?). If published, this will include your full peer review and any attached files.

Reviewer #1: No

Reviewer #2: No

Reviewer #3: No

**Data Deposition**

http://datadryad.org/submit?journalID=pgenetics&manu=PGENETICS-D-25-00416R1

**Press Queries**

---

## [Editor Report · Acceptance letter]

PGENETICS-D-25-00416R1

Statistical methods to disentangle genetic effects influencing infertility and early fetal viability with a genome-wide application

Dear Dr Skodvin,

We are pleased to inform you that your manuscript entitled " 

Statistical methods to disentangle genetic effects influencing infertility and early fetal viability with a genome-wide application" has been formally accepted for publication in PLOS Genetics! Your manuscript is now with our production department and you will be notified of the publication date in due course.

With kind regards,

Anita Estes

PLOS Genetics

On behalf of:
